# Anisotropy and Frequency Dependence of Signal Propagation in the Cerebellar Circuit Revealed by High-Density Multielectrode Array Recordings

**DOI:** 10.3390/biomedicines11051475

**Published:** 2023-05-18

**Authors:** Anita Monteverdi, Danila Di Domenico, Egidio D’Angelo, Lisa Mapelli

**Affiliations:** 1Brain Connectivity Center, IRCCS Mondino Foundation, 27100 Pavia, Italy; 2Department of Brain and Behavioral Sciences, University of Pavia, 27100 Pavia, Italy

**Keywords:** cerebellum, multielectrode arrays, granule cells, Purkinje cells, electrophysiology

## Abstract

The cerebellum is one of the most connected structures of the central nervous system and receives inputs over an extended frequency range. Nevertheless, the frequency dependence of cerebellar cortical processing remains elusive. In this work, we characterized cerebellar cortex responsiveness to mossy fibers activation at different frequencies and reconstructed the spread of activity in the sagittal and coronal planes of acute mouse cerebellar slices using a high-throughput high-density multielectrode array (HD-MEA). The enhanced spatiotemporal resolution of HD-MEA revealed the frequency dependence and spatial anisotropy of cerebellar activation. Mossy fiber inputs reached the Purkinje cell layer even at the lowest frequencies, but the efficiency of transmission increased at higher frequencies. These properties, which are likely to descend from the topographic organization of local inhibition, intrinsic electroresponsiveness, and short-term synaptic plasticity, are critical elements that have to be taken into consideration to define the computational properties of the cerebellar cortex and its pathological alterations.

## 1. Introduction

The cerebellar network displays a well-organized modular architecture repeating itself almost identically along the entire cerebellar cortex [1]. Briefly, inputs are provided by climbing fibers (from the inferior olive, directly contacting Purkinje cells) and mossy fibers (from the rest of the brain), which send collaterals to the deep cerebellar nuclei before reaching the cortex. Here, mossy fibers (MFs) contact granular layer neurons (inhibitory Golgi cells and the most abundant neurons in the entire brain, the excitatory granule cells) in anatomical structures called “glomeruli” [2]. The granular layer is therefore considered the input layer of the cerebellar cortex. In a complex network of inhibitory loops [3], the signals are conveyed to Purkinje cells (PCs), which are the output layer of the cortex. Molecular layer interneurons (MLI) are contacted by granule cells (GrCs) and inhibit PCs. The anatomical orientation of most of these neurons influences their function. Golgi cells (GoCs) develop the axonal plexus in the sagittal plane [4,5], While the parallel fibers, i.e., GrC axons providing excitation to MLI and PCs, are organized orthogonally with respect to this plane (i.e., in the coronal plane). PCs develop their wide dendritic arborization in the sagittal plane and can also be activated by GrC ascending axons (before generating the parallel fibers; see Figure 1). MLI axons contact PCs both in the sagittal (mainly basket cells) and coronal (mainly stellate cells) planes [6,7,8].

This uniform and ordered network processes inputs in an extremely complex way, and a comprehensive characterization of cerebellar functions and network dynamics currently represents an open challenge. The recently recognized role of the cerebellum in higher cognitive and emotional processes, besides the well-studied sensorimotor integration, makes it even more crucial to address the issue [9,10,11,12,13,14].

Indeed, the cerebellum operates in different frequency ranges, receiving extensive and heterogeneous inputs from the motor and non-motor areas, and the cerebellar cortical activity is known to be considerably frequency dependent. The granular layer appears to be well equipped for developing and maintaining rhythmic low-frequency activity (theta band, 2–10 Hz), presenting coherent oscillations and resonance [15]. In the 20 Hz input domain, an increase in the responsiveness of GrCs and GoCs and a coherent organization of the granular layer emergent activity has been predicted [16]. At higher frequencies (50–100 Hz), MFs stimulation has been described to maximally evoke the response of PCs located over the excited granular layer area [17]. Recently, a nonlinear frequency dependence of the complex mechanisms regulating neurovascular coupling in the cerebellum has been characterized [18]. Short-term dynamics, through depression and facilitation of synaptic transmission, are known to be powerful modulators of signal transmissions in neuronal networks [6,19,20,21]. At the mossy fiber–granule cell relay, short-term depression is known to enhance the inhibitory control of synaptic gain, allowing multiplicative operations of the inputs by the postsynaptic neurons [22]. At the same time, the PC output depends on the combination of the short-term plasticity (mainly potentiation) at the parallel fiber connection, and that with MLI, both deeply frequency dependent [20,23]. This explains the diversity of PC response to mossy fiber stimulation, ranging from net increases in the basal firing to complete stops of spontaneous firing. Through these mechanisms, short-term plasticity is able to impact the excitatory/inhibitory balance of the granular and PC layers [20] and modulate the information transfer through the whole network [19]. However, to date, the experimental assessment of granular and PC spikes responses to a varying range of mossy fiber inputs in the sagittal and coronal planes was still missing, and a complete understanding of the frequency dependence of cerebellar cortical processing is yet to be achieved.

In this work, we characterized the frequency dependence of cerebellar processing and its short-term dynamics in the cortical input and output layers both in the sagittal and coronal orientations. We have taken advantage of a high-density multielectrode array (HD-MEA) device, a recent cutting-edge technique that allows simultaneous extracellular recordings from 4096 electrodes, sampling the activity from all the cerebellar cortical layers. Network responses to MFs stimulation at different frequencies (6 Hz, 20 Hz, 50 Hz, 100 Hz), mimicking different ranges of brain activity, were analyzed and compared in two different planes of section (sagittal and coronal) in acute mouse cerebellar slices. Our results integrate previous investigations on cerebellar frequency dependence [17] and, thanks to technological advancement, reveal an unforeseen efficiency of the cerebellar cortical network in transmitting low-frequency inputs. Moreover, cerebellar short-term dynamics showed a marked orientation dependence, revealing anisotropic signal transmission compatible with the anatomical organization of local inhibitory circuits.

## 2. Materials and Methods

Animal maintenance and experimental procedures were performed according to the international guidelines of the European Union Directive 2010/63/EU on the ethical use of animals and were approved by the local ethical committee of the University of Pavia (Italy) and by the Italian Ministry of Health (authorization following art. 1, comma 4 of the D. Lgs. n. 26/2014 approved on 9 December 2017).

### 2.1. Slice Preparation and Maintenance

Acute parasagittal or coronal cerebellar slices (220 µm thick) were obtained from 18 to 23-day-old C57BL/6 mice of either sex. Animals were anesthetized with halothane (Aldrich, Milwaukee, WI, USA) and killed by decapitation. The cerebellum was gently removed to isolate the vermis, fixed with cyanoacrilic glue on the specimen support of a vibroslicer (VT1200S, Leica Microsystems, Wetzlar, Germany), used to obtain the slices. The whole procedure was performed in cold and oxygenated Krebs solution which contained (in mM): 120 NaCl, 2 KCl, 1.2 MgSO4, 26 NaHCO3, 1.2 KH2PO4, 2 CaCl2, 11 glucose, pH 7.4 equilibrated with 95% O_2_–5% CO_2_. Slices were recovered for 1 hour in the same solution before recording. To record cerebellar activity, slices were gently positioned on the HD-MEA chip improving the coupling with the electrodes array using a platinum ring with a nylon mesh. Oxygenated Krebs solution (2–3 mL/min) was continuously perfused in the glass reservoir during the whole recording session and maintained at 32 °C with a Peltier feedback temperature controller (TC-324B; Warner Instrument Corporation, Holliston, MA, USA).

### 2.2. High-Resolution Electrophysiological Recordings

The investigation of the cerebellar circuit was conducted using a complementary metal-oxide-semiconductor (CMOS) based high-density multielectrode array (HD-MEA; Biocam X, 3Brain AG, Wädenswil, Switzerland). The chip consisted of 4096 electrodes arranged in a 64 × 64 matrix on an area of 2.67 mm × 2.67 mm. Electrode size was 21 μm × 21 μm with a pitch of 42 μm (Biochip Arena, 3Brain AG, Wädenswil, Switzerland). The whole chip is packaged onto a substrate together with a glass reservoir with a diameter of 25 mm and 7 mm of height. Signals were sampled at 18 kHz/electrode, with a high pass filter at 100 Hz. Electrical stimulation was provided using a bipolar tungsten electrode positioned on the MFs bundle (pulses of 50 μA with a duration of 200 μs per pulse). This stimulation setting is demonstrated to selectively activate MFs, without direct GrC excitation [24]. To investigate cerebellar input processing at different frequencies, 30 single pulses were delivered at 0.1 Hz, followed by 30 trains of 5 impulses at 6 Hz, 20 Hz, 50 Hz, and 100 Hz (in mixed combinations) repeated every 10 s.

### 2.3. Data Analysis

#### 2.3.1. Local Field Potentials

MFs stimulation elicited GrCs response observed in the form of Local Field Potentials (LFPs) propagating through the granular layer. The LFP was characterized by a typical N_1_-N_2a_-N_2b_-P_2_ complex. In this complex, N1 derives from the activation of the presynaptic volley, N_2a,_ and N_2b_ are informative of GrCs synaptic activation, and P2 represents the current returning from the molecular layer [18,25]. Data were displayed online and stored using the BrainWave X Software (3Brain AG, Wädenswil, Switzerland) and data analysis was performed using ad-hoc routines written in MATLAB (Mathworks). To investigate GrCs synaptic activation, only N_2a_ and N_2b_ peaks were considered for the analysis. Peak amplitudes were calculated by subtracting the negative peaks found in the appropriate time window (1.9 ± 1.1 ms and 4.5 ± 1.2 ms from the stimulus artifact, for N_2a_ and N_2b_, respectively; [25]) to the baseline derived from the averaged signal in 300 ms before the stimulus onset. LFP signals were considered for the analysis only when the peak amplitude exceeded 3 times (for N_2a_) and 2.5 times (for N_2b_) the standard deviation calculated over the baseline period, in response to at least 75% of trials. LFP signals stability (see right inset in Figure 2) over the entire duration of the experiment was verified using an unpaired Student’s *t*-test. Channels showing a significant increase or decrease of the N_2a_ peak amplitude in response to single pulse stimulation or to the first stimulation in the pulse trains at the end of the recording were discarded (significance at *p* < 0.05). The percent change of the last response in the train, compared to the first one, within each stimulation, pattern was calculated both for N_2a_ and N_2b_, and then used to generate colormaps to represent the corresponding spatial distribution. The signals showing a statistically significant change (unpaired Student’s *t*-test, *p* < 0.05) in N_2a_ and N_2b_ peak amplitudes at the end of the 5 pulses stimulation compared to the first response in the train were considered undergoing short-term plasticity. Data are reported as mean ± SEM (standard error of the mean).

#### 2.3.2. Purkinje Cell Firing

PCs autorhythmic spiking activity was recorded using the BrainWave X software (3Brain AG, Wädenswil, Switzerland). The analysis of PCs firing was performed using BrainWave 4 software (3Brain AG, Wädenswil, Switzerland) and ad-hoc routines written in MATLAB (Mathworks). PCs were identified based on three main parameters: (i) their location on the slice, between the granular and molecular layers following the lobule layout; (ii) their firing frequency, ranging from 10 to more than 100 Hz; (iii) the spike amplitude exceeding 100 μV. Possible contamination by basket cells cannot be completely ruled out, given that these cells have their cell bodies located in the internal molecular layer and some can be found near PCs. However, basket cells are smaller than PCs and usually have lower spontaneous discharge frequency, so their spikes are unlikely to be detected and confused with those of the PCs. Therefore, although we cannot exclude that some basket cells could have been included in the analysis, their impact on the results should be negligible. Moreover, complex spikes were never observed after MF stimulation but could be elicited by positioning the stimulating electrode underneath the PC layer (see left inset in Figure 2). Spike detection of PC activity was performed in BrainWave 4 using the hard threshold of −100 μV (refractory period of 1 ms). Spike detection was followed by the waveform extraction in a temporal window of 0.5 ms pre-spike and 3 ms post-spike and the spike sorting using a Principal Component Analysis based on spikes characteristics followed by a clustering k-mean algorithm. Pakhira-Bandyopadhyay-Maulik (PBM) index was used for validating clustering results. Due to the inter-electrode tip distance of 42 μm and the large PC soma (25–40 μm diameter), commonly more than a unit was detected by every single electrode of the HD-MEA. The BrainWave 4 software was used to eliminate common units in nearby channels and retain the signal only in the channel presenting spikes with the largest amplitude. In this way, the same PC was not considered multiple times in different electrodes. The user’s supervision of every step of this procedure was performed. Then peri-stimulus time histograms (PSTHs) and raster plots were used for the analysis of PCs’ responses to stimulation. PC responses consisted of a transient increase or decrease of the basal discharge. Herein, an increase in PC firing frequency was defined as a peak in the PSTH, while a decrease in firing frequency was defined as a pause in the PSTH. Peaks and pauses detection was performed using one-tailed permutation test [26]. The permutation test was restricted to increased Mean Firing Rate (MFR) for peaks and decreased MFR for pauses. In both cases, it was followed by a false discovery rate correction with alpha at 0.01. PSTHs resolution was set at 5 ms/bin to analyze peaks and 20 ms/bin to analyze pauses. For each PC detected, the basal MFR was calculated on the 500 ms pre-stimulus period and used to generate colormaps. The percent change of the MFR within each stimulation pattern was calculated by comparing the MFR at the end (during the last two pulses) of the trains at different input frequencies to the MFR at the beginning (during the first two pulses) of the trains at different input frequencies. This percent change was used to generate colormaps together with the percent change of N_2a_ peak amplitude obtained from the previous analysis. Data are reported as mean ± SEM.

## 3. Results

The activity of the granular and PC layers was recorded using a high-density multielectrode array (HD-MEA), which provides an exceptional spatiotemporal resolution [18]. In both sagittal and coronal slices, MFs were stimulated to activate the cerebellar cortical network. As evident in Figure 1, MFs contact both GrCs and GoCs in the granular layer. GrCs axons originate the parallel fibers, which contact PCs, MLIs, and GoCs. GrCs are then inhibited through feedforward (MF–GoC–GrC) and feedback (GrC–GoC–GrC) loops. In turn, PCs are excited by GrCs and inhibited by MLIs, which provide inhibitory loops in the molecular layer. It is evident that the inhibitory component is predominant both in the granular and molecular layers, to determine cerebellar cortical processing, [2,3]. The architecture of the cerebellar cortex is such that forcing a sagittal or coronal orientation to the circuit activation (by using the slicing procedure) is expected to impact cerebellar processing. In this work, we characterized how cerebellar input and output layers responses are affected, at different activity ranges, by the sagittal or coronal orientation of the circuit.
Figure 1The cerebellar circuit. (**A**) 3D orthographic view of the cerebellar network. As can be seen, mossy fibers (MF) convey an excitatory input in the granular layer (GL), contacting granule cells (GrC) and Golgi cells (GoC). GoC develop their axonal plexus in the sagittal plane and exert a feedforward and a feedback inhibition onto GrC. GrC axon (aa) passes vertically the Purkinje cells layer (PCL) and reaches the molecular layer (ML) originating parallel fibers (PF). These fibers are organized orthogonally in the coronal plane and make excitatory synapses onto Purkinje cells (PC). PC develop their dendritic arborization in the sagittal plane and their activity is under the inhibitory control exerted by stellate cells (SC) and basket cells (BC). Dendrites and axons are of different color grades (lighter and darker, respectively). (**B**) Schematic representation of the cerebellar cortical circuit. Forward and stop arrows indicate excitatory and inhibitory connections, respectively.
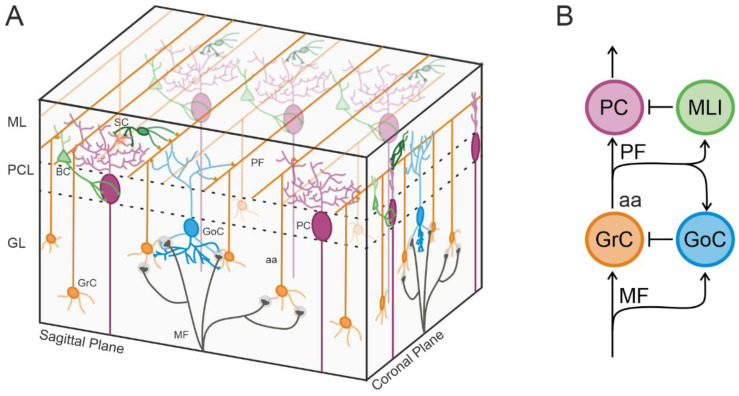



### 3.1. Characterization of Cerebellar Cortical Activity with HD-MEA

HD-MEA recordings are the ideal tool to access the spontaneous activity of PCs and the responses to MFs stimulation on both the granular layer (silent at rest) and the PC layer itself (Figure 2). PCs spontaneous activity was detected by 1859 channels in 9 sagittal slices and 1809 channels in 9 coronal slices, resulting in the characterization of 858 and 1095 single units, respectively (Table 1; see Methods for details on the spike sorting procedure). The basal MFR of the detected units was 58.1 ± 2.8 Hz in sagittal slices (*n* = 9) and 85.5 ± 8.0 Hz in coronal slices (*n* = 9; Student’s *t*-test, *p* = 0.001). PCs showed heterogeneous basal MFR ranging from 10 Hz to more than 100 Hz. A color map representing the spatial organization of PCs with different basal MFR was reconstructed (Figure 3A). Interestingly, in coronal slices PCs with low and high basal MFR tended to alternate in bands, while in sagittal slices no evident pattern could be detected. Though this kind of characterization is beyond the scope of this work, this spatial organization resembles the zebrin-like distribution pattern described for PCs in the coronal plane [27].

Single-pulse MFs stimulation evoked GrCs responses, recorded as LFPs propagating through the granular layer of the stimulated lobule, both in sagittal and coronal slices (Figure 2). The LFP showed the typical N1-N_2a_-N_2b_-P2 complex in agreement with previous observations [18,24,25]. Herein, the analysis was focused on N_2a_ and N_2b_ peaks as a measure of postsynaptic GrCs activation (see Methods for details). Over the large sample obtained (804 channels in 9 sagittal slices and 1649 channels in 9 coronal slices; Table 1), the average peak delays and amplitudes for N_2a_ and N_2b_ peaks were as follows. N_2a_: 1.60 ± 0.02 ms and −233.3 ± 21.4 μV in parasagittal slices; 1.55 ± 0.04 ms and −214.0 ± 19.6 μV in coronal slices. N_2b_: 4.51 ± 0.09 ms and −76.8 ± 3.8 μV in parasagittal slices; 4.25 ± 0.16 ms and −71.8 ± 1.9 μV in coronal slices. N_2a_ and N_2b_ peaks delays and amplitudes did not present a significant difference in sagittal and coronal slices (unpaired Student’s *t*-test, N_2a_ *p* = 0.32 and *p* = 0.51, N_2b_ *p* = 0.19 and *p* = 0.26). The LFP spread through the granular layer was recorded along the stimulated lobule, allowing us to compute the propagation velocity of the signal. In sagittal and coronal slices, signal velocity was respectively 0.66 ± 0.09 m/s and 0.44 ± 0.09 m/s, not presenting significant differences (unpaired Student’s *t*-test, *p* = 0.12).

The table shows the number of channels in nine parasagittal and nine coronal slices detecting local field potential (LFP) signals in the granular layer and Purkinje cells firing (ch = channels). The number of units detected in sagittal and coronal slices after spike detection and spike sorting operations is reported. The last column contains the total amount of channels and units analyzed from all cerebellar slices.
Figure 2Typical HD-MEA recording. (**A**) For both sagittal (upper panels) and coronal (lower panels) orientations, the images in the center show the cerebellar slice on the chip (scale bar 250 μm), with the stimulating electrode (black line) positioned on the MFs. The colored dots over the slice represent the single HD-MEA channels showing neuronal activity in the selected time-bin (500 ms at left, 3 ms at right). The raw traces represent examples of recordings of PCs’ spontaneous activity (left) and local field potential (LFP, right). The N_2a_ and N_2b_ peaks of the LFP are indicated in the electrophysiological trace. The plot in the inset shows the time course of the average N_2a_ peak amplitude in response to single pulse stimulation and to the first stimulation in the trains, for all the channels used for the analysis. (**B**) Schematic representation of the stimulation protocol.
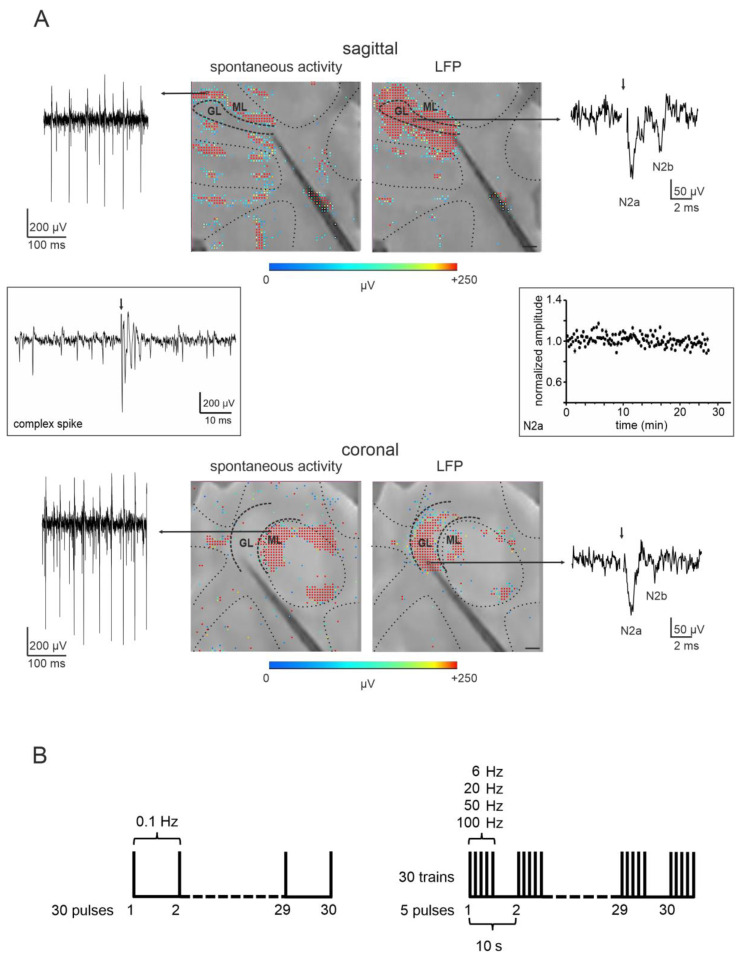



To characterize PC responses, raster plots, and PSTHs were reconstructed for each PC detected (Figure 3B,C). Over a total of 1953 units (858 in sagittal slices and 1095 in coronal), two different types of responses were identified: a rapid significant increase of PCs basal firing in the PSTH (within 20 ms after the pulse, here defined as “peak”) and a significant decrease of PCs firing in the PSTH (within 40 ms after the pulse, here defined as “pause”). As evident in Figure 3B, PCs’ most common response pattern was an increase in their basal discharge after a single pulse stimulation in sagittal slices and a decrease in their basal discharge in coronal slices.

For simplicity, the detailed analysis of cerebellar cortical responses to different frequencies of MFs stimulation is reported separately for the experiments performed in sagittal and coronal slices. The comparison between these two conditions is highlighted in a devoted paragraph at the end of Section 3.
Figure 3PCs’ spontaneous and evoked activity. (**A**) The pictures show the cerebellar slice on the chip, with superimposed the location of the analyzed channels for PCs’ spontaneous activity, for a sagittal (left) and coronal (right) experiment. For each channel, the basal mean firing rate (MFR) of a single unit is represented using the color scale on the right. Scale bar 200 μm. (**B**) The histograms show the percentage of PCs showing peaks or pauses in response to single-pulse MF stimulation in sagittal and coronal slices (** *p* < 0.01). (**C**) Examples of raster plots and PSTHs of PCs’ activity are reported for both sagittal (left) and coronal (right) slices. The PSTH shows the two main classes of PCs’ responses: a significant increase of PCs firing in the PSTH (5 ms-bin, peak) or a significant decrease of PCs firing in the PSTH (20 ms-bin, pause). The PSTH in the time window corresponding to −0.5/0.5 s is magnified below. Scale bar 0.5 sp/bin.
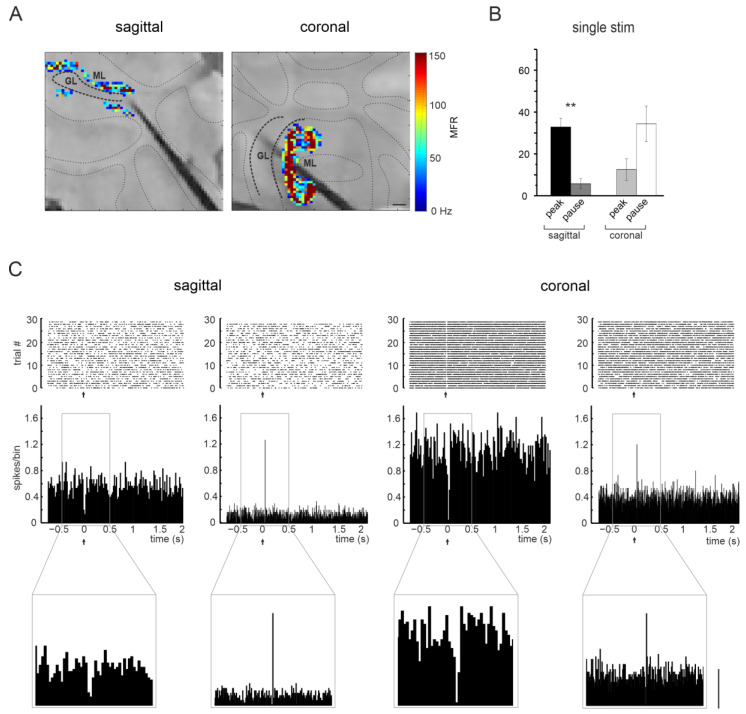



### 3.2. Short-Term Plasticity on the Sagittal Plane

#### 3.2.1. Granular Layer Responses at Different Input Frequencies

To characterize granular layer responses to different ranges of activity, MFs were stimulated with 5 pulses at 6, 20, 50, and 100 Hz. The amplitude of the LFP N_2a_ and N_2b_ peaks following each pulse was measured and normalized for comparison (Figure 4). 

The percent change of the amplitude in response to the last pulse in the train, compared to the first one, was used to determine the direction of the short-term plasticity recorded. The results showed a trend to decrease for N_2a_ peak amplitude, while the N_2b_ peak showed significantly more variability (paired Student’s *t*-test, *p* = 0.009) at increasing frequency, increasing or decreasing its amplitude during the stimulation trains (Table 2 and Figure 4). However, while the percentage of channels showing a significant change in N_2a_ peak amplitude was always high in the granular layer (more than 70% of the recorded channels), the percentage of channels detecting a significant change in N_2b_ was more than 40% only at high input frequencies (Table 2).

On average, N_2a_ peak amplitude showed short-term depression, becoming more evident increasing the stimulation frequency. In turn, N_2b_ appeared to weakly tend to increase especially at higher input frequencies, but the heterogeneous profile evident in the channel population needs to be considered (Figure 4).

#### 3.2.2. The Spatial Organization of Short-Term Plasticity in the Granular Layer

The spatial resolution of the HD-MEA was exploited to investigate the spatial organization of granular layer responses in the stimulated lobules. The previously calculated percent changes of N_2a_ and N_2b_ peaks following MFs stimulation at the different frequencies tested were used to generate color maps showing the position of each channel (and the relative response) in the cerebellar lobule. Colormaps constructed with N_2a_ peak percent change suggested a spatial organization of short-term plasticity in the granular layer (Figure 5A). Larger short-term depression appeared to be more pronounced in the center of the granular layer, becoming more evident at increasing stimulation frequencies. An average colormap of all the recordings was reconstructed aligning the slices along the MF axis, showing a sort of longitudinal spatial organization of short-term plasticity in the sagittal plane (Figure 5B).

The spatial organization of N_2b_ percent changes in the granular layer appeared to be different than N_2a_. As evident in Figure 5A, the trend to increase the N_2b_ peak appeared more accentuated in the center of the granular layer, and this phenomenon was more evident increasing the input frequency. The reconstruction of the average color map (Figure 5B) showed that this spatial organization is consistent in the different slices, though less defined in the longitudinal axes at all frequencies compared to the one shown for N_2a_.

#### 3.2.3. Purkinje Cells Responses at Different Input Frequencies

PCs responses were evaluated at all input frequencies (6 Hz, 20 Hz, 50 Hz, 100 Hz). Unfortunately, the short time interval between pulses at 100 Hz hampered the reconstruction of PC responses within the train at that frequency. The results are reported as a percentage of units presenting MFR increases or decreases in response to the stimulation. As reported in Figure 6A, the PCs’ most common response pattern was an increase (PSTH peak) in the basal discharge after each stimulation pulse at all the frequencies examined, while the decrease in the basal discharge (PSTH pause) was less common (see statistics in Figure 6A). This trend was more evident at increasing frequencies.

Finally, the percent change of the MFR during the stimulation train was calculated within each stimulation pattern (6 Hz, 20 Hz, 50 Hz, excluding the 100 Hz case for the above-mentioned limitation). Similar to what was already described for the granular layer, the comparison between the last and the first response in the stimulation train was calculated, here as percent changes in the MFR. A special representation of the units was obtained generating colormaps showing the percent change. These maps were combined with the ones generated for the granular layer to describe the short-term modifications in neuronal activity in both layers of the cerebellar cortex (Figure 6B). Notice that increasing the input frequency determined a marked decrease in granular N_2a_ amplitude while a marked increase in PCs MFR.
Figure 6PCs responses to MF stimulation in the sagittal plane. (**A**) The histograms show the average percentage of units responding to MF stimulation at different frequencies with peaks (black) or pauses (grey) in the PSTHs. Asterisks indicate statistical significance (* *p* < 0.05, ** *p* < 0.01, *** *p* < 0.001). (**B**) Colormaps showing the percent change of PCs MFR at the last stimulation pulse compared to the first one, at 6, 20, and 50 Hz, in a single experiment (scale bar 100μm). In the same image, the channels in the granular layer showing a response to MF stimulation are reported, for the same experiment, showing the percent change of the N_2a_ peak (as in Figure 5A) using the same color code at right.
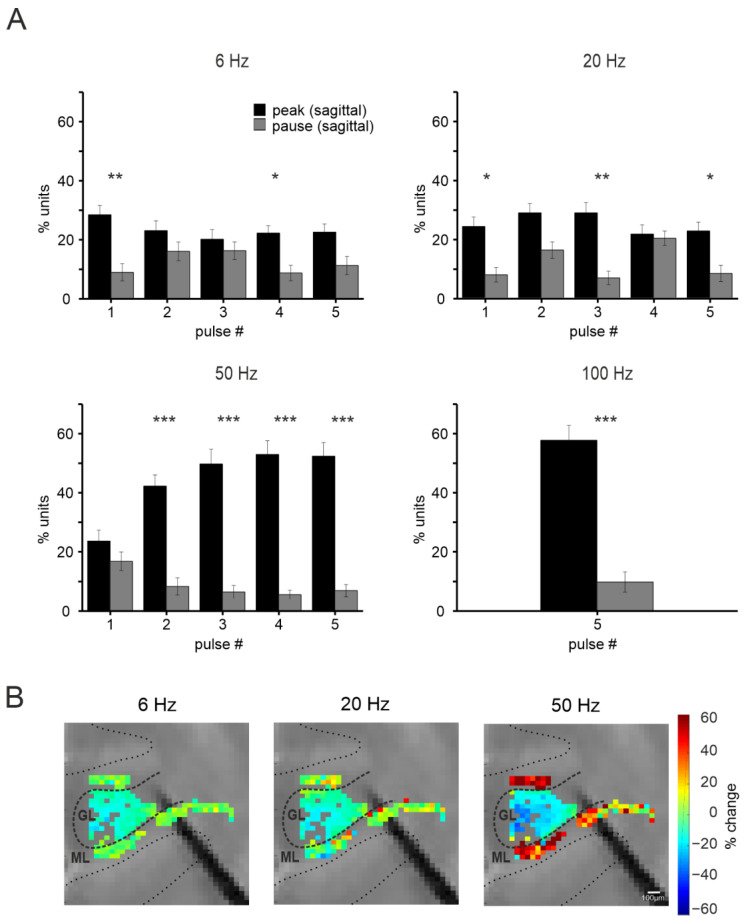



### 3.3. Short-Term Plasticity on the Coronal Plane

#### 3.3.1. Granular Layer Responses at Different Input Frequencies

The same analysis performed on sagittal slices was repeated on 9 coronal slices. The percent change of N_2a_ and N_2b_ peaks was calculated at different input frequencies together with the percentage of channels showing short-term plasticity of N_2a_ and N_2b_ peak amplitudes in the granular layer. Again, the decrease of N_2a_ peak amplitude prevailed within the stimulation trains, becoming more evident increasing the stimulation frequency (Figure 7).

N_2b_ showed both an increase and a decrease in its amplitude (Table 3), but its variability during the train was not significantly increased (paired Student’s *t*-test *p* = 0.35) at increasing frequency. Moreover, among the channels showing a response to the stimulation, the percentage of channels showing short-term plasticity in the N_2a_ peak amplitude was above 50% at every condition, while it reached 40% only at higher input frequencies (50 Hz and 100 Hz) for the N_2b_ peak (Table 3). The percentage of channels showing a significant increase in the N_2b_ peak amplitude was larger than the percentage of channels presenting a decrease (Table 3).

N_2a_ and N_2b_ peak amplitude changes for each stimulus in the trains are reported in Figure 7, showing an average predominant short-term depression for N_2a_ peak amplitude and an average weak increase for N_2b_ peak amplitude. Again, the heterogeneous profile evident in the channel population when considering N_2b_ peak amplitude needs to be considered.

#### 3.3.2. The Spatial Organization of Short-Term Plasticity in the Granular Layer

The percent change of N_2a_ and N_2b_ peaks in each recorded channel was used to construct colormaps representing the spatial organization of the responses in the granular layer. Again, short-term depression of the N_2a_ peak was more pronounced in the center of the responding region. This was more evident at increasing frequencies, suggesting a frequency dependence (Figure 8A). The average colormap was reconstructed aligning all the recordings along the MF axis, confirming the organization observed in each single experiment (Figure 8B).

The N_2b_ peak amplitude changes during the stimulation showed a less defined spatial organization, compared to N_2a_ (Figure 8A). Nevertheless, the increase in the N_2b_ peak was more concentrated in the center of the responding region nearest to the stimulation site. This was more evident in the average colormaps. Again, this effect was enhanced at increasing stimulation frequencies (Figure 8B).
Figure 8The spatial organization of short-term plasticity in the coronal plane. (**A**) Color maps were reconstructed for each stimulation pattern with the percent change of N_2a_ (top panels) and N_2b_ (bottom panels) peak amplitude after the last stimulation pulse compared to the first one, from a single experiment. Scale bar 100 μm. (**B**) Average colormaps obtained aligning the recorded slices along the MFs axis, showing the average percent change of N_2a_ (top panels) and N_2b_ (bottom panels) peak amplitude. The white cross indicates the stimulus location.
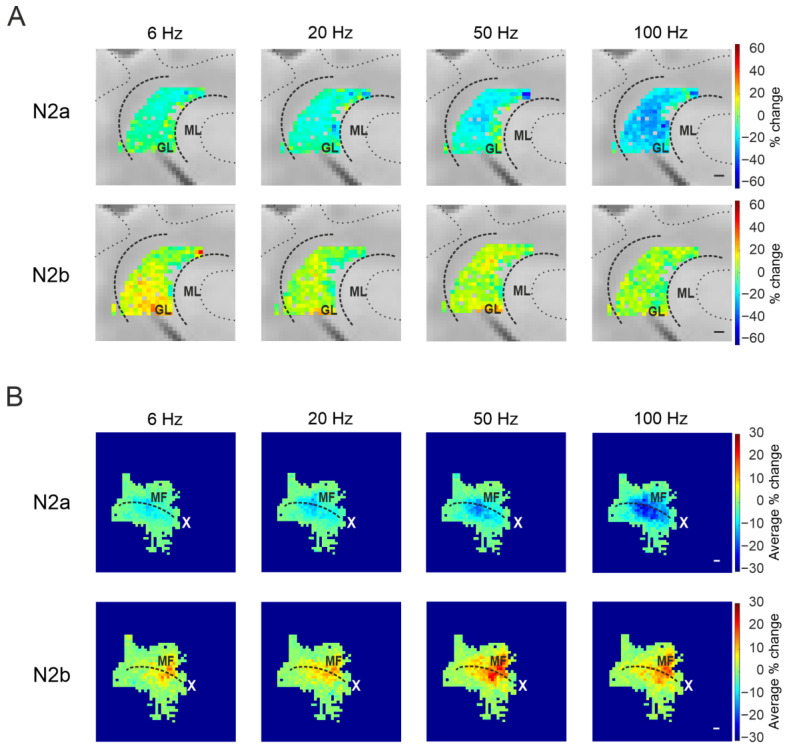



#### 3.3.3. Purkinje Cells Responses at Different Input Frequencies

Raster plots and PSTHs reconstructed for each stimulation frequency showed a combination of peaks and pauses over a total of 1095 units. During stimulation at different frequencies, PC responses showed pauses prevailing at the beginning and peaks prevailing at the end of the five-pulse train (Figure 9A).

Finally, the percent change of the MFR was calculated within each stimulation pattern (6 Hz, 20 Hz, 50 Hz), excluding the 100 Hz for the limitations specified above. Colormaps were generated using this percent change and combined with the ones describing granular layer short-term plasticity, for each experiment (Figure 9B), representing the spatio-temporal reconstruction of cerebellar cortical responses at different input frequencies in the coronal plane. Notice that increasing the input frequency determined a marked decrease in granular N_2a_ amplitude while a marked increase in PCs MFR.
Figure 9PCs responses to MF stimulation in the coronal plane. (**A**) The histograms show the average percentage of units responding to MFs stimulation at different frequencies with peaks (black) or pauses (grey) in the PSTHs. Asterisks indicate statistical significance (* *p* < 0.05, ** *p* < 0.01). (**B**) Colormaps showing the percent change of PCs’ mean firing rate (MFR) at the last stimulation pulse compared to the first one, at 6, 20, and 50 Hz, in a single experiment. In the same image, the channels in the granular layer showing a response to MFs stimulation are reported, for the same experiment, showing the percent change of the N_2a_ peak (as in Figure 5A) using the same color code at right. Scale bar 100 μm.
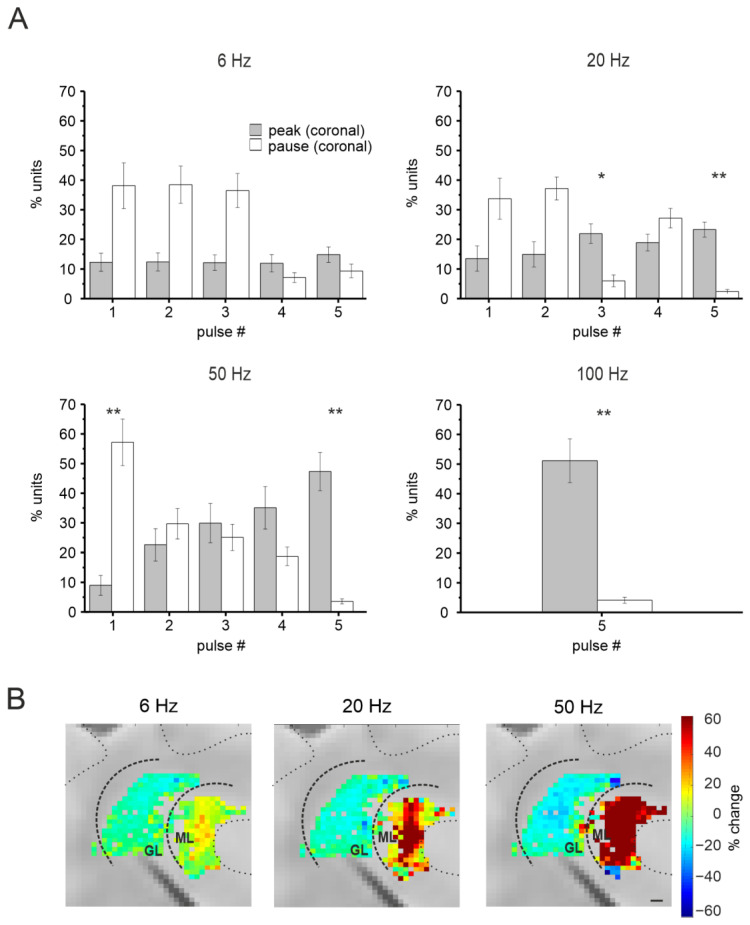



### 3.4. Comparison of Cerebellar Network Responses in the Sagittal and Coronal Planes

MFs stimulation evoked an LFP response propagating through the granular layer. The activated granular layer area appeared to be more extended in coronal slices than in sagittal ones. After a single pulse stimulation, N_2a_ and N_2b_ peaks characterizing GrCs responses did not show a statistically significant difference in the two planes of the section. However, at increasing input frequencies, N_2b_ peak amplitude changes during the train appeared to differ in the two conditions, concerning short-term depression. Indeed, while N_2a_ showed short-term depression in both sagittal and coronal slices with similar amplitude, the N_2b_ peak could show an increase or a decrease in its amplitude. The main differences comparing the responses in the sagittal and coronal planes can be detected at higher frequencies (50 Hz and 100 Hz) for the amount of short-term depression shown (at 50 Hz, much smaller in the coronal plane; *p* = 0.001) and the percentage of channels showing depression (at 50 Hz and 100 Hz, much smaller in the coronal plane; *p* = 0.01, *p* = 0.03). This is in agreement with the fact that, at these frequencies, N_2b_ showed short-term potentiation in the coronal plane more than in the sagittal one (*p* = 0.04) (Figure 10A).

Concerning PCs activity, their average basal MFR was significantly higher in coronal than in sagittal slices. As discussed below, this might depend on the different composition in zebrin positive and negative cells in the coronal and sagittal slices, and on different orientations of MLI axons. Following MFs stimulation, PCs showed marked differences in their response patterns already after a single pulse. While peak responses significantly prevailed in the sagittal plane, pause responses were much more common in the coronal. This difference was conserved in the responses within the trains. In both sagittal and coronal slices, peaks prevailed over pauses at the end of the 5 pulses at any frequency, though more evidently at 50 Hz and 100 Hz. Nevertheless, PCs in the coronal plane maintained a higher degree of pause responses within the train compared to the sagittal plane, where pauses were less represented at any frequency (cfr. Figure 6 and Figure 9). Concerning PC response amplitude, the MFR percent change within each stimulation pattern was not different in sagittal and coronal slices (Figure 10B).

## 4. Discussion

In this work, last-generation HD-MEA allowed us to characterize cerebellar microcircuit processing with state-of-the-art resolution. HD-MEA provided LFPs recording in the granular layer and PC single-unit spikes over the surface extension of a cerebellar slice, which allowed us to reconstruct the spatiotemporal distribution of activity in response to input patterns at different frequencies. The main observation is that cerebellar network processing is markedly anisotropic and frequency dependent.

Mossy fiber signal transmission through the cerebellar microcircuit was observed over an extended frequency range (0.1–100 Hz) and increased with frequency. Interestingly, we observed a previously underestimated efficiency of the network in processing low-frequency inputs. Moreover, while previous investigations suggested that only the sagittal orientation had an impact on dynamic processing, with a relative frequency independence in the coronal plane [17], here we show that both planes are characterized by an evident frequency dependence. Yet another aspect is that of short-term processing dynamics, which differed between the sagittal and coronal orientations, with an impact on the network output more evident at lower frequencies. All these differences presumably reflect the topographic organization of local inhibition, intrinsic electroresponsiveness, and short-term synaptic plasticity. After a few considerations on HD-MEA recordings, the results are briefly summarized to facilitate the reader in following the logical development of the Discussion leading to the schematic reconstruction of signal transmission summarized in Conclusions.

### 4.1. Considerations on HD-MEA Recordings

The HD-MEA allowed us to record the granular layer responses from an average of 136 channels per slice (2453 in 18 slices; see Table 1). Considering that 10–15 GrCs are expected to contribute to the LFP recorded in a single channel in these conditions (see [18]), the response detected in a single slice might be provided by approximately 1700 GrCs. However, we cannot exclude that a single GrC contributed to the signal recorded in two nearby channels, though the probability that the contribution was significant is low given the small cell size and the locality of the electrical field generated around it [28]. The average GrC diameter is around 5 µm [29], while the pitch between the electrodes is 42µm. Moreover, the electrode is recessed by about 1.5 µm in the chip insulating layer, restricting the sensitivity to the above neurons. Concerning PCs, the average number of detected units was 108 per slice (1953 in 18 slices; see Table 1). In this case, a single PC may indeed contribute to several nearby channels. For this reason, an automated procedure was used to avoid considering the same PC unit more than once (see Methods for details). Indeed, the average number of channels detecting PC activity was 204 per slice (3668 in 18 slices). This high throughput allowed us to test a large set of input patterns with high spatiotemporal resolution.

A comparison of responses in sagittal and coronal cerebellar slices was also performed. The anatomy of the cerebellar cortical circuit is geometrically orientated: in the granular layer, GoC inhibition over GrCs is exerted more efficiently in the sagittal plane (given the specific orientation of the GoC axonal bundle); in the molecular layer, excitatory parallel fibers travel in the coronal plane, intersecting PCs and MLI dendrites that are oriented parasagittally. As previously reported [17], thanks to this anatomical organization of the circuit, the use of sagittal and coronal slices is an effective tool to discriminate signal transmission through the two main orientation planes of the cerebellar cortex. Thus, the HD-MEA allowed us a straightforward investigation of the anisotropic activation of GrCs and PCs.

This technique allowed us to resolve spatiotemporal network dynamics with enhanced resolution compared to low-density MEA [25] or voltage-sensitive dye imaging [17]. Low-density MEAs had less than half the HD-MEA pitch and a lower signal-to-noise ratio. Voltage-sensitive dyes allowed a good spatial resolution (<10 μm), but this was diffraction-limited and the quantum yield of the dyes was rather low decreasing the signal-to-noise ratio. Moreover, voltage-sensitive dyes were incapable of resolving single spikes and required a signal average. It should also be noted that HD-MEA signals mainly depend on neuronal firing, both in the granular and Purkinje cell layer, while voltage-sensitive dye imaging informs both about subthreshold and suprathreshold membrane potential changes.

Taken together, these features need to be considered when comparing the present data to previous ones obtained using different techniques.

### 4.2. Characterization of Spontaneous and Evoked Activity in Granular and PC Layers

GrCs are silent at rest and constitute the majority of cells in the granular layer with a ratio of about 400:1 with respect to GoCs in rodents [18,29,30], while PCs are autorhythmic with a mean firing rate (MFR) ranging from 20 to more than 100 Hz [31]. Interestingly, our data show that PCs’ basal MFR in the coronal is significantly higher than in the sagittal plane. Several factors might be involved. First, parallel fibers are intact in the coronal plane, increasing the glutamatergic tone on PCs. Secondly, basket cell axons target PCs in the sagittal plane, so that the inhibitory tone on PC spontaneous firing should be lower in coronal slices, where basket cell axons are severed. Another factor that might have contributed to different PC firing rates in coronal and sagittal slices is zebrin protein expression. Zebrin-positive PCs are reported to have a lower spontaneous firing rate than zebrin-negative PCs [32] (from 36 Hz to 76 Hz in Ref. [33]) The zebrin pattern develops in parasagittal stripes and we do not know whether positive or negative PCs were equally represented in sagittal and coronal slices.

MF stimulation determined responses both in the granular and PC layer. The granular layer response was characterized by an LFP showing a series of peaks [24]. Interestingly, the number of channels showing GrCs responses was larger in the coronal plane. This might depend on the anisotropy of MF rosettes distribution in the granular layer, being three times closer one to the other in the coronal plane compared to the sagittal [34]. An increase in rosettes density might recruit more GrCs and, therefore, result in LFP detection in a larger number of channels. In granular layer LFPs, N_2a_ reflects the synchrony of GrCs firing driven by AMPA receptor-mediated responses, while N_2b_ reflects the NMDA receptor-mediated component of GrC synaptic responses and is modulated by Golgi cell GABAergic inhibition [25,28]. No difference between sagittal and coronal planes was found in the delay and amplitude of N_2a_ and N_2b_. Conversely, PC responses revealed a significant difference. In the sagittal plane, most PCs showed a transient increase in their firing rate (evident as a peak in the PSTH). (Figure 3B). In the coronal plane, most PCs showed a pause after the stimulation (Figure 3B). The anatomy of the circuit provides hints for the interpretation of these data. In the sagittal plane, PCs are mostly activated by GrCs ascending axons, while the inhibition provided by MLIs is weak (GrC → PCs). The functional equivalence of ascending axons and parallel fiber inputs has been demonstrated [35]. MLIs connections are more preserved in the coronal plane, as for parallel fibers. Therefore, GrCs activation by MFs is likely to activate MLIs through parallel fibers and lead to the consequent inhibition of PCs (parallel fiber → MLI 
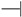
 PC pathway).

### 4.3. Frequency-Dependent Responses in the Granular and PC Layers

The granular and PC layer responses to stimulation trains at different frequencies reveal a complex scenario.

The granular layer shows N_2a_ short-term depression that increases with frequency, reflecting the short-term depression of the AMPA receptor-mediated component of MF-GrCs responses [36,37,38,39,40,41]. The spatial organization of N_2a_ short-term depression could be due to the different number of MFs impinging on GrCs at different locations (more abundant along the medial region of the lobule). An increased number of active MFs contacts would amplify GrC AMPARs desensitization. Conversely, the N_2b_ component of the LFP can show either a decrease or an increase. This duality of response changes during the trains probably reflects the different excitatory/inhibitory balance that finally regulates the extent of NMDA receptor activation [2,18,37,42]. N_2b_ variability was significantly increased at an increasing frequency only in the sagittal plane, while N_2b_ depression at 50 Hz was smaller in the coronal orientation. This may reflect the anatomical organization of the GoC axonal plexus, which lies in the sagittal plane [3,4,5], exerting more effective control over GrCs firing in the sagittal than the coronal plane.

PCs responses differ in the two orientations at lower frequencies, showing mainly an MFR increase in sagittal and MFR decrease in the coronal plane. PCs responses at higher frequencies is a net MFR increase in both orientations. The PC response pattern in the coronal plane is richer than in the sagittal, being characterized by pauses at the beginning of the stimulation trains and peaks at the end. The frequency dependence of MLIs inhibition [6] and the short-term dynamics shaping signal transmission and MLIs activity [20] are likely to contribute to this shift. Indeed, different types of short-term plasticity have been observed at parallel fibers synapses depending on the target neuron, with short-term facilitation at parallel fibers–stellate cells relay and short-term depression at parallel fibers–basket cells relay [43,44]. Parallel fibers–MLIs synapses can be characterized by heterogeneous profiles of short-term potentiation, determining differences in the firing frequencies and delays of MLIs responses at different input frequencies [45]. Since complex spikes were not detected in response to MF stimulation, climbing fibers’ contribution to MLIs, and PC activity [46] in our experiments can be excluded. In aggregate, PCs patterns of response in the coronal plane are most likely compound in their origin, and they probably involve a stronger drive from the granular layer in the coronal plane, determining the shift from pause to peak responses observed during prolonged high-frequency stimulation, and determining the loss of the anisotropy typical of low-frequency transmission.

## 5. Summary and Conclusions

A complex set of spatiotemporal filters regulates signal transmission in the cerebellar cortex, revealing a marked frequency dependence and anisotropy.

In the granular layer, the N_2a_ wave of the LFP always shows increasing depression with frequency, while N_2b_ shows either potentiation or depression depending on the granular layer microdomain. Therefore, N_2b_ can discriminate the bandpass at the input stage. When N_2b_ depresses, the filter sharpens, becomes less permissive at higher frequencies, and decreases the time window for GrC firing (this case is more common on the sagittal plane). When N_2b_ potentiates, the filter dampens, decreases its impact at higher frequencies, and allows longer GrC discharge (this case is more common on the coronal plane). Since N_2b_ depends on GABAergic control over NMDA receptor unblocking, this differential behavior reveals an important role for GoC inhibition in determining granular layer microdomains with diverse filtering properties. PCs add to the richness of signal transmission properties in the network. At low frequencies, PCs show either an increase or decrease in firing, with a clear orientation preference (sagittal or coronal, respectively). Nonetheless, at high frequencies, all PCs show a sharp increase in firing.

In aggregate, on the sagittal plane, the GrC time window for emitting spikes is sharp and PCs increase firing already at low frequencies. On the coronal plane, the GrC time window for emitting spikes is broad and PCs decrease firing at low frequencies (Figure 11). It is possible that the increased time window of GrC firing on the coronal plane determines the response properties of PCs in synergy with frequency-dependent MLI control. Testing this hypothesis will require computational modeling [47] opening new perspectives for the calculation of network complexity, entropy, and mutual information transfer.

## Figures and Tables

**Figure 4 biomedicines-11-01475-f004:**
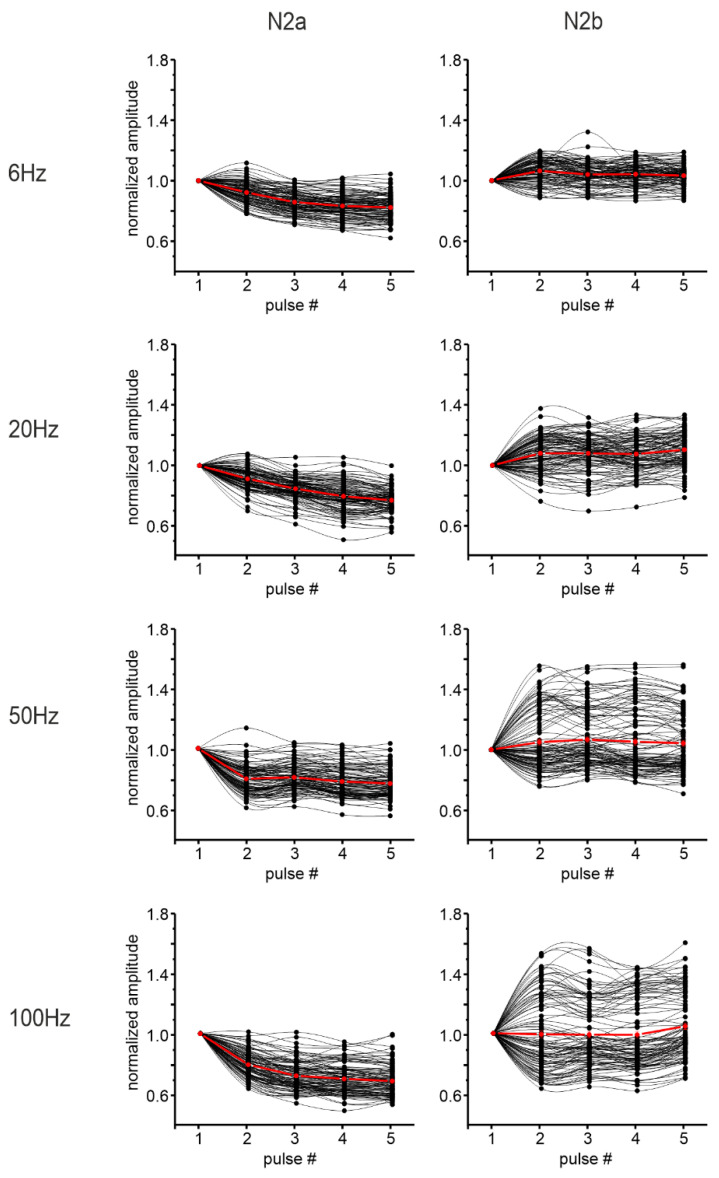
Granular layer responses to stimulation trains in sagittal slices. The plots show the normalized N_2a_ (left column) and N_2b_ (right column) peak amplitudes within the stimulation trains at the different frequencies tested. The responses to each of the five stimulation pulses in the train were normalized to the response to the first pulse. Black dots represent the single channels considered for the analysis. The red dots show the average trend.

**Figure 5 biomedicines-11-01475-f005:**
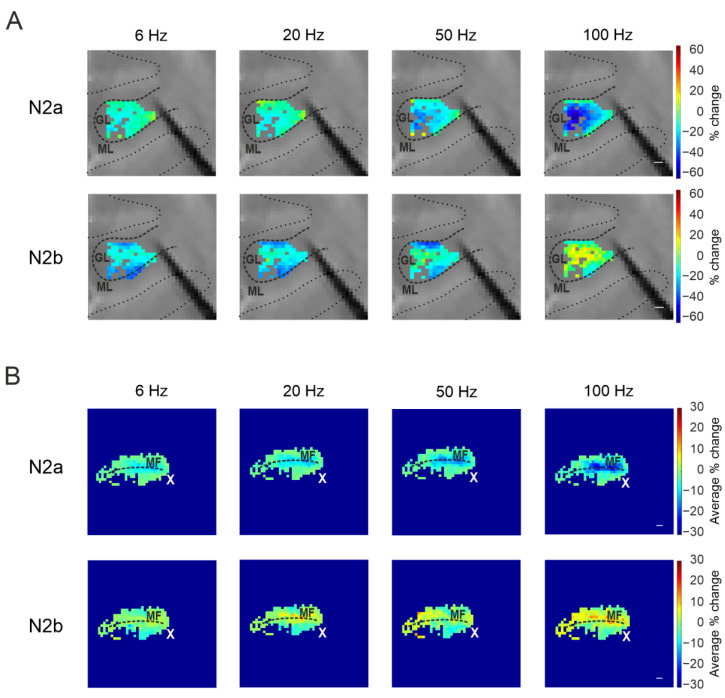
The spatial organization of short-term plasticity in the sagittal plane. (**A**) Color maps were reconstructed for each stimulation pattern with the percent change of N_2a_ (top panels) and N_2b_ (bottom panels) peak amplitude after the last stimulation pulse compared to the first one, from a single experiment. Scale bar 100 μm. (**B**) Average colormaps obtained aligning the recorded slices along the MFs axis, showing the average percent change of N_2a_ (top panels) and N_2b_ (bottom panels) peak amplitude. The white cross indicates the stimulus location.

**Figure 7 biomedicines-11-01475-f007:**
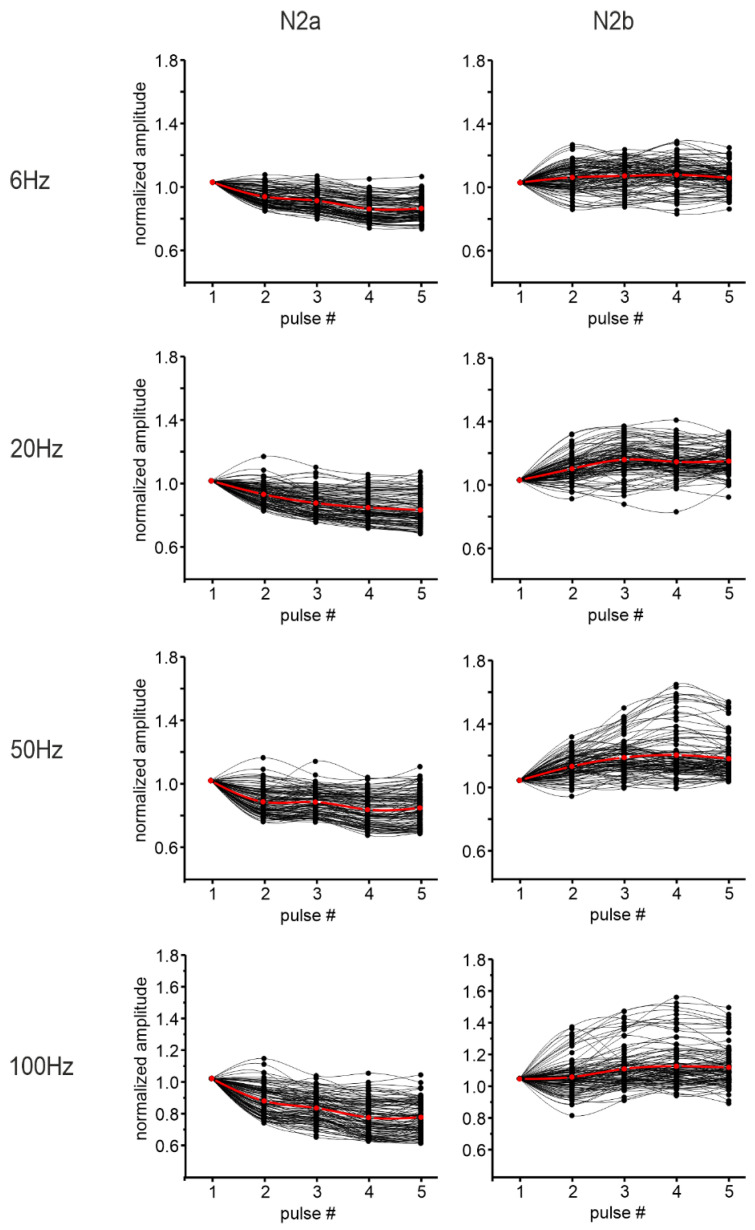
Granular layer responses to stimulation trains in coronal slices. The plots show the normalized N_2a_ (left column) and N_2b_ (right column) peak amplitudes within the stimulation trains at the different frequencies tested. The responses to each of the five stimulation pulses in the train were normalized to the response to the first pulse. Black dots represent the single channels considered for the analysis. The red dots show the average trend.

**Figure 10 biomedicines-11-01475-f010:**
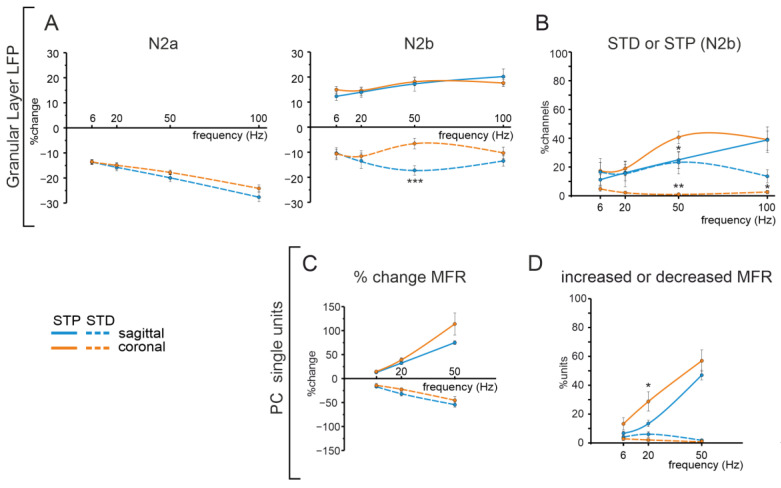
Comparison of the granular layer and PC responses in the sagittal and coronal planes. (**A**) The plots show the average percent change of the granular layer N2a (left) and N2b (right) peak amplitude in the sagittal (blue) and coronal orange planes. (**B**) The plot shows the percentage of units showing a significant change of N2b peak amplitude (as short-term potentiation, STP, full line; and short-term depression, STD, dashed line) in sagittal (blue) and coronal orange slices. (**C**) The plot shows the average percent change of PC mean firing rate (MFR) in sagittal (blue) and coronal orange slices. (**D**) The plot shows the percentage of units showing a significant increase (full line) and decrease (dashed line) of PCs MFR after the last stimulation pulse at different frequencies in the sagittal (blue) and coronal orange planes. Asterisks indicate statistical significance between granular layer and PC responses (either as STP or STD) in sagittal and coronal orientations (* *p* < 0.05, ** *p* < 0.01, *** *p* < 0.001).

**Figure 11 biomedicines-11-01475-f011:**
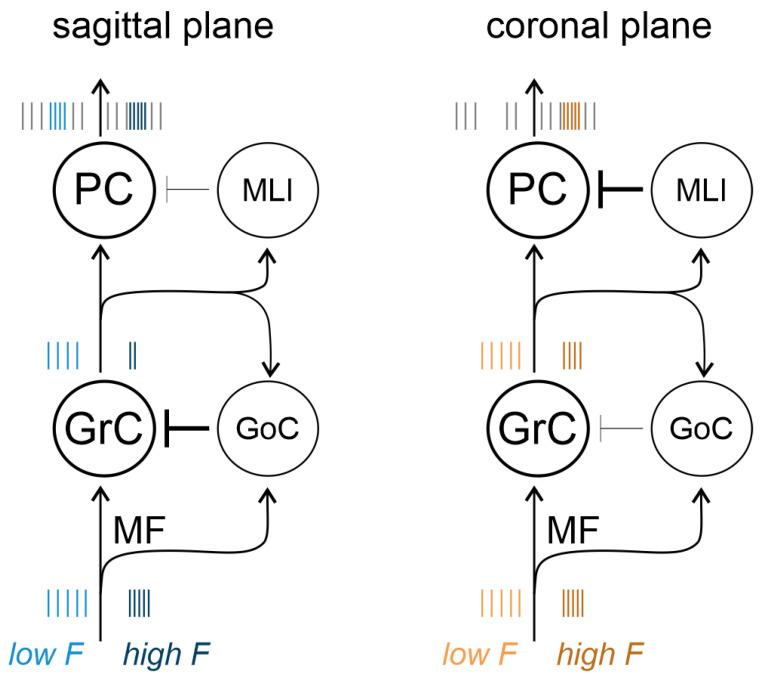
Schematic representation of low and high-frequency signal transmission on the sagittal and coronal planes. Left: on the sagittal plane, granule cells (GrC) receive robust inhibition from Golgi cells (GoC) and Purkinje cells (PC) receive weak inhibition from molecular layer interneurons (MLI). The granular layer filtering of MF input sharpens at high frequency. Right: on the coronal plane, GoC inhibition of GrC activity is less prominent and MLI control of PC activity is more robust. The time window for GrC firing broadens and PC show a decrease in firing at low frequency and an increase in firing at high frequency.

**Table 1 biomedicines-11-01475-t001:** Recording channels and detected units.

	SAGITTAL	CORONAL	
	*n* = 9	*n* = 9	tot
ch granular layer	804	1649	2453
ch Purkinje cells	1859	1809	3668
detected units	858	1095	1953

**Table 2 biomedicines-11-01475-t002:** N_2a_ and N_2b_ peak amplitude changes at different input frequencies (sagittal plane). (**A**) The table shows the percent change of the peak amplitude in response to the last pulse compared to the first one within each stimulation pattern. (**B**) The table shows the percentage of channels in the granular layer undergoing N_2a_ or N_2b_ short-term plasticity.

A				
	**6 Hz**	**20 Hz**	**50 Hz**	**100 Hz**
N_2a_	−13.60 ± 0.99	−15.58 ± 1.45	−19.87 ± 1.41	−27.37 ± 1.92
N_2b_ **↑**	12.33 ± 1.68	13.89 ± 2.01	17.13 ± 2.83	20.09 ± 3.18
N_2b_ **↓**	−10.59 ± 2.42	−13.36 ± 3.08	−17.34 ± 1.85	−13.48 ± 2.10
B				
	**6 Hz**	**20 Hz**	**50 Hz**	**100 Hz**
%ch N_2a_	75.82 ± 4.63	72.34 ± 6.48	83.92 ± 3.85	88.82 ± 3.03
%ch N_2b_ ↑	11.83 ± 3.61	16.94 ± 5.74	25.74 ± 5.88	40.09 ± 9.05
%ch N_2b_ ↓	16.57 ± 9.41	15.15 ± 8.86	22.81 ± 7.61	13.56 ± 4.49

In both tables, N_2b_ ↑ stands for N_2b_ amplitude increase during the stimulation trains and N_2b_ ↓ refers to N_2b_ amplitude decrease during the stimulation trains.

**Table 3 biomedicines-11-01475-t003:** N_2a_ and N_2b_ peak amplitude changes at different input frequencies (coronal plane). (**A**) The table shows the percent change of the last response peak compared to the first one within each stimulation pattern. (**B**) The table shows the percentage of channels in the granular layer recording a significant N_2a_ or N_2b_ peak amplitude change.

A				
	**6 Hz**	**20 Hz**	**50 Hz**	**100 Hz**
N_2a_	−13.44 ± 0.84	−14.92 ± 1.26	−17.56 ± 0.89	−23.93 ± 1.46
N_2b_ +	14.80 ± 1.44	14.26 ± 1.65	18.23 ± 1.69	17.62 ± 1.41
N_2b_ −	−10.63 ± 1.71	−11.55 ± 2.19	−6.53 ± 2.12	−10.28 ± 2.30
B				
	**6 Hz**	**20 Hz**	**50 Hz**	**100 Hz**
%ch N_2a_	53.96 ± 7.18	55.47 ± 6.85	61.59 ± 6.73	78.94 ± 5.37
%ch N_2b_ ↑	18.01 ± 6.02	19.77 ± 4.94	41.60 ± 4.43	39.41 ± 6.78
%ch N_2b_ ↓	4.75 ± 1.60	2.23 ± 0.79	0.74 ± 0.29	2.60 ± 1.12

In both these tables N_2b_ ↑ stands for N_2b_ amplitude increase during the stimulation trains while N_2b_ ↓ refers to N_2b_ amplitude decrease during the stimulation trains.

## Data Availability

All the materials related to the paper are available from the corresponding author upon reasonable request.

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
