# Peer review of "Anisotropy and Frequency Dependence of Signal Propagation in the Cerebellar Circuit Revealed by High-Density Multielectrode Array Recordings"

_biomedicines, 2023, doi:10.3390/biomedicines11051475_

Round 1

Reviewer 1 Report

The manuscript by Monteverdi et al is about the analysis of the responsiveness of cerebellar networks to electrical stimulation of mossy fiber inputs using high density multielectrode array. The approach followed by the authors is very interesting and provide new data regarding the processing of synaptic stimulation via the mossy fiber pathway. Experiments and analysis are well done and results are well described. The manuscript is well written. I have only few comments I would like the authors to addressed to improve the reading for non-specialists and would help to translate the approach to other brain structures.

In figure 1, the connectivity is not well shown and there is a confusion between dendrite and axon and which is presynaptic and postosynaptic. Authors may add a simplified schematic showing input and output of each cells type and reciprocal connectivity, including the inhibitory-excitatory nature of the connection.

In all figures, the images showing slices are not clear and it is quite challenging to figure out the orientation and the layer organization. It is difficult to recognize a cerebellar slice.

In legend of Figure 2, the sentence “the colored pixels…” is not clear.

Authors claim they stimulate mossy fibers and then stimulate granule cells. However, how the authors can be sure they don’t stimulate directly granule cells?

In figure 4, the variability of the N2b response during the train increases with the stimulation frequency. Can the author make a comment?

Author Response

The manuscript by Monteverdi et al is about the analysis of the responsiveness of cerebellar networks to electrical stimulation of mossy fiber inputs using high density multielectrode array. The approach followed by the authors is very interesting and provide new data regarding the processing of synaptic stimulation via the mossy fiber pathway. Experiments and analysis are well done and results are well described. The manuscript is well written. I have only few comments I would like the authors to addressed to improve the reading for non-specialists and would help to translate the approach to other brain structures.

We thank the Reviewer for the positive comments. We have addressed each comment, as explained in the point-by-point reply below.

In figure 1, the connectivity is not well shown and there is a confusion between dendrite and axon and which is presynaptic and postosynaptic. Authors may add a simplified schematic showing input and output of each cells type and reciprocal connectivity, including the inhibitory-excitatory nature of the connection.

Figure 1 has been modified as suggested. Dendrites and axons are of different color grade, as now specified in the legend. Moreover, a scheme of the connectivity has been added in panel B, to clarify input and output of each neuronal type and the inhibitory or excitatory nature of the connection.

In all figures, the images showing slices are not clear and it is quite challenging to figure out the orientation and the layer organization. It is difficult to recognize a cerebellar slice.

The outline of the lobules has been indicated on all figures containing slice images.

In legend of Figure 2, the sentence “the colored pixels…” is not clear.

We thank the Reviewer for noticing this indication, which was indeed confusing. The legend has been modified with “the colored dots over the slice”, indicating the MEA channels showing neuronal activity in the represented time bin, as further reported in the legend.

Authors claim they stimulate mossy fibers and then stimulate granule cells. However, how the authors can be sure they don’t stimulate directly granule cells?

Mossy fibers stimulation using a bipolar tungsten electrode is a common procedure in cerebellar research and has been used by this and other laboratories over the last twenty years. A thorough characterization of the effect of this stimulation is reported in Maffei et al., 2002, where specific experiments were conducted to elucidate whether other excitable elements could be directly excited in this condition. These experiments demonstrated that neither granule cells nor Purkinje cells were directly excited by the stimulating electrode. Granule cells were involved in the synaptic response to mossy fiber stimulation. This is evident also in our experiments, since granular layer responses have an average delay of about 1.6ms, compatible with synaptic transmission and not with direct electrical stimulation. The reference Maffei et al. (2002) is now reported in the Methods section with a sentence on the specificity of this stimulation protocol (line 116).

In figure 4, the variability of the N2b response during the train increases with the stimulation frequency. Can the author make a comment?

We thank the Reviewer for pointing out this variability. Indeed, the N2b standard deviation is significantly increased at high frequency but only in sagittal slices. This evidence of anisotropic activation of the cerebellar circuit is likely to reflect the impact of Golgi cells in shaping granule cells activity, which is more efficient in the sagittal than coronal plane. The corresponding statistics are now reported in Results (lines 301, 400) and commented in Discussion (line 642).

Reviewer 2 Report

General comments:

In this study, the authors used a high-throughput high-density multielectrode array to compare the spread of stimulus-evoked activity in the sagittal and coronal planes of acute mouse cerebellar slices. They observed frequency dependence and spatial anisotropy of the cerebellar activation, as summarized in Fig. 11. The presented data may provide new insights into cerebellar physiology. But there remains a serious concern that the observed anisotropy may not be a physiological but an artificial one due to the differential disruptions of inhibitory neurons in the sagittal and coronal slices. Namely, axons of molecular layer interneurons are damaged more severely in sagittal slices. In contrast, axons of Golgi cells are damaged more severely in coronal slices, as suggested by the authors themselves. It is necessary to provide more convincing evidence that the former is the case. Alternatively, it may be possible to use the difference as experimental manipulation in the discussion.  

Major comments:

1.     Page 3, line 115: “a bipolar tungsten electrode positioned on the MFs bundle” It doesn’t necessarily mean you stimulated MFs selectively.

2.     Page 3, line 148: “Purkinje cell firing”  You should describe how you identified PCs more thoroughly.  

3.     Page 4, lines 169-171: “In order to optimize PSTHs resolution, 5ms bin width was used to analyze peaks and 20ms bin width was chosen for pauses”.  Why are these different bin widths optimal?

4.     Page 10, lines 366-368: “Notice that increasing the input frequency determined a marked decrease in granular N2a amplitude while a marked increase in PCs MFR”.  Discussing the functional interpretation of these observations in Discussion would be best.

5.     Page 14, line 449: “the percent change of the MFR was calculated within each stimulation pattern (6Hz, 20Hz, 50Hz)”  On line 171 (page 4), you described that “the MFR was calculated on the 500ms prestimulus”. I cannot understand how you calculate MFR for 5 pulses that last less than 500ms at 6Hz, 20Hz, and 50Hz. You should provide a better explanation.

6.     Page 15, lines 461-462: “B) Colormaps showing the percent change of PCs mean firing rate (MFR) at the last stimulation pulse compared to the first one, at 6, 20, and 50Hz, in a single experiment”.  It is even more puzzling how you can calculate MFR (500ms window?) for each stimulus pulse.

7.     Figure 9B: Why is the distribution of PCs in the coronal slice much wider than in the sagittal slice (e.g., Fig. 6)? It suggests that the coronal slice significantly deviated from a coronal plane. The granular layer in the coronal slice also appears much wider than in the sagittal slice.

8.     Page 17, Discussion:  Discussion is a summary of Results. I appreciate a more active discussion on your observation.

Minor comments:

1.     Page 3, lines 116-118: “To investigate cerebellar input processing at different frequencies, 30 single pulses were delivered at 0.1 Hz, followed by 30 trains of 5 impulses at 6Hz, 20Hz, 50Hz, and 100Hz (in mixed combinations) repeated every 10 seconds”.  It is beneficial to provide a figure to explain the stimulus protocol. What does “mixed combination” mean? Did you test these different frequencies in a randomized order?

2.     Page 3, line 135: “seen inset in Fig. 1” -> see inset in Fig. 2

3.     Page 4, lines 171-172: “pre-stimulus” -> pre-stimulus period

4.     Page 6, line 246: “positioned on the MFs”  You cannot exclude stimulation of other neuronal elements.

5.     Page 7, Figure 3B: Why do not the histograms sum up to 100%?

6.     Page 7, Figure 3 C: Recognizing peaks and pauses in the four panels is difficult. How about using a more limited time window, e.g., from -0.6s to +0.6s?

7.     Page 10, Figure 5B: The red crosses are too small.

Author Response

General comments:

In this study, the authors used a high-throughput high-density multielectrode array to compare the spread of stimulus-evoked activity in the sagittal and coronal planes of acute mouse cerebellar slices. They observed frequency dependence and spatial anisotropy of the cerebellar activation, as summarized in Fig. 11. The presented data may provide new insights into cerebellar physiology. But there remains a serious concern that the observed anisotropy may not be a physiological but an artificial one due to the differential disruptions of inhibitory neurons in the sagittal and coronal slices. Namely, axons of molecular layer interneurons are damaged more severely in sagittal slices. In contrast, axons of Golgi cells are damaged more severely in coronal slices, as suggested by the authors themselves. It is necessary to provide more convincing evidence that the former is the case. Alternatively, it may be possible to use the difference as experimental manipulation in the discussion. 

The cerebellar cortical circuit anatomical orientation is well known and is supposed to influence signal transmission. As reported in Introduction: “The anatomical orientation of most of these neurons influences their function. Golgi cells (GoCs) develop the axonal plexus in the sagittal plane [4,5]. while the parallel fibers, i.e., GrC axons providing excitation to MLI and PCs, are organized orthogonally with respect to this plane (i.e., in the coronal plane). PCs develop their wide dendritic arborization in the sagittal plane and can also be activated by GrC ascending axons (before generating the parallel fibers; see Fig.1). MLI axons contact PCs both in the sagittal (mainly basket cells) and coronal (mainly stellate cells) planes [6–8].” This anatomical orientation is exploited here by using sagittal and coronal cutting planes that allowed us to investigate the anisotropic spread of activity in the circuit. In detail, the comparison between sagittal and coronal slices allowed us to better understand the role of Golgi cell inhibition in shaping granule cells responses as well as the role of the parallel fiber – molecular layer interneurons – Purkinje cell feedforward loop in determining Purkinje cell inhibition. Therefore, the two cutting planes helped us to understand the physiological implications of the anisotropic anatomical orientation of the circuit. A previous example of the efficacy of this methodology can be found in Mapelli et al., 2010, where a similar approach was applied to voltage-sensitive dye recordings in cerebellar slices. A comment on this issue and this reference are now reported in Discussion (section 4.1, line 574).

Major comments:

  1. Page 3, line 115: “a bipolar tungsten electrode positioned on the MFs bundle” It doesn’t necessarily mean you stimulated MFs selectively.

Mossy fibers stimulation using a bipolar tungsten electrode is a common procedure in cerebellar research and has been used by this and other laboratories over the last twenty years. A thorough characterization of the effect of this stimulation is reported in Maffei et al., 2002, where specific experiments were conducted to elucidate whether other excitable elements could be directly excited in this condition. These experiments demonstrated that neither granule cells nor Purkinje cells were directly excited by the stimulating electrode. Granule cells were involved in the synaptic response to mossy fiber stimulation. This is evident also in our experiments, since granular layer responses have an average delay of about 1.6ms, compatible with synaptic transmission and not with direct electrical stimulation. The reference Maffei et al. (2002) is now reported in the Methods section with a sentence on the specificity of this stimulation protocol (line 116).

Page 3, line 148: “Purkinje cell firing”  You should describe how you identified PCs more thoroughly.

A further specification has been added to Methods: “PCs were identified based on three main parameters: i) their location on the slice, between the granular and molecular layers following the lobule layout; ii) their firing frequency, ranging from 10 to more than 100Hz; iii) the spike amplitude exceeding 100μV.”

  1. Page 4, lines 169-171: “In order to optimize PSTHs resolution, 5ms bin width was used to analyze peaks and 20ms bin width was chosen for pauses”. Why are these different bin widths optimal?

We thank the Reviewer for pointing to the term “optimize”. Indeed, we did not perform a mathematical process of optimization to determine the bin size. We simply relied on our experience and used the bin size that allowed to discriminate the response, as in previous works (Ramakrishnan et al., 2016; Moscato et al, 2019). Therefore, we changed the text as follows: “PSTHs resolution was set at 5ms/bin to analyze peaks and 20ms/bin to analyze pauses”.

Refs:

  Ramakrishnan KB, Voges K, De Propris L, De Zeeuw CI, D'Angelo E. (2016) Tactile Stimulation Evokes Long-Lasting Potentiation of Purkinje Cell Discharge In Vivo. Front Cell Neurosci. doi: 10.3389/fncel.2016.00036. PMID: 26924961

  Moscato L, Montagna I, De Propris L, Tritto S, Mapelli L, D'Angelo E. (2019) Long-Lasting Response Changes in Deep Cerebellar Nuclei in vivo Correlate With Low-Frequency Oscillations. Front Cell Neurosci. doi: 10.3389/fncel.2019.00084. PMID: 30894802

  1. Page 10, lines 366-368: “Notice that increasing the input frequency determined a marked decrease in granular N2a amplitude while a marked increase in PCs MFR”. Discussing the functional interpretation of these observations in Discussion would be best.

This issue is discussed in the Conclusions and summarized in Fig.11.

  1. Page 14, line 449: “the percent change of the MFR was calculated within each stimulation pattern (6Hz, 20Hz, 50Hz)” On line 171 (page 4), you described that “the MFR was calculated on the 500ms prestimulus”. I cannot understand how you calculate MFR for 5 pulses that last less than 500ms at 6Hz, 20Hz, and 50Hz. You should provide a better explanation.

The MFR calculated on the 500ms prestimulus describes the basal firing frequency of that PC. To avoid confusion, this has now been renamed “basal MFR”, to distinguish it from the MFR measured at the beginning and end of the stimulation train and used to calculate the percent change. These procedures are now clarified in the text.

  1. Page 15, lines 461-462: “B) Colormaps showing the percent change of PCs mean firing rate (MFR) at the last stimulation pulse compared to the first one, at 6, 20, and 50Hz, in a single experiment”. It is even more puzzling how you can calculate MFR (500ms window?) for each stimulus pulse.

The clarification about point 4 also applies here.

  1. Figure 9B: Why is the distribution of PCs in the coronal slice much wider than in the sagittal slice (e.g., Fig. 6)? It suggests that the coronal slice significantly deviated from a coronal plane. The granular layer in the coronal slice also appears much wider than in the sagittal slice.

The three-dimensional organization of cerebellar lobules is such that, in the coronal plane, some slices show more PC lines. This depends on the specific location of the cutting section: if it coincides with lobule bending, PCs are distributed over in multiple lines.

  1. Page 17, Discussion: Discussion is a summary of Results. I appreciate a more active discussion on your observation.

Given the complex set of results reported in this paper, we preferred to briefly summarize the data before discussing them. This is meant to facilitate the reader in following the logical development of Discussion. We added a sentence at the beginning of Discussion to clarify the presentation strategy: “After a few considerations on HD-MEA recordings, the results are briefly summarized to facilitate the reader in following the logical development of Discussion leading to the schematic reconstruction of signal transmission summarized in Fig.11.”

Minor comments:

  1. Page 3, lines 116-118: “To investigate cerebellar input processing at different frequencies, 30 single pulses were delivered at 0.1 Hz, followed by 30 trains of 5 impulses at 6Hz, 20Hz, 50Hz, and 100Hz (in mixed combinations) repeated every 10 seconds”. It is beneficial to provide a figure to explain the stimulus protocol. What does “mixed combination” mean? Did you test these different frequencies in a randomized order?

We added a panel in Fig.2 to represent the stimulation protocol. We chose to avoid the term “random” since it refers to a precise mathematical procedure. Anyway, we changed the order of the tested frequency at each experiment and controlled whether each order induced plastic changes (see stability in Methods and Fig.2). Therefore, we preferred the use of the term “mixed combination”.

  1. Page 3, line 135: “seen inset in Fig. 1” -> see inset in Fig. 2

Thank you for noticing, we changed it in the text.

  1. Page 4, lines 171-172: “pre-stimulus” -> pre-stimulus period

We changed the text accordingly.

  1. Page 6, line 246: “positioned on the MFs” You cannot exclude stimulation of other neuronal elements.

The clarification at point 1 also answers to this concern.

  1. Page 7, Figure 3B: Why do not the histograms sum up to 100%?

The histograms show the percentage of Purkinje cells showing peaks or pauses in response to single pulse MF stimulation in sagittal and coronal slices. Histograms do not sum up to 100% since not all PCs responded to single pulse stimulation.

  1. Page 7, Figure 3 C: Recognizing peaks and pauses in the four panels is difficult. How about using a more limited time window, e.g., from -0.6s to +0.6s?

We magnified the region around the response below the PSTH reported in Fig.3, as suggested.

  1. Page 10, Figure 5B: The red crosses are too small.

We increased the size and changed the color of the crosses, to improve visibility.

Reviewer 3 Report

In this article, Monteverdi and colleagues investigated the spread of neuronal activity across the cerebellar cortex induced by the stimulation of mossy fibers in the granule cell layer. The analysis of the spatial distribution of single unit recordings using high-throughput high-density multielectrode array revealed an anisotropic and frequency-dependent propagation of the signal with interesting differences according to the orientation of the slice (sagittal versus coronal). The results suggest a role for local inhibition and short-term plasticity in controlling the propagation of the signal, which should impact how cerebellar microcircuits process information.

Overall, the article is logically written and the methodological approach is sound. The results bring new and interesting information that will be of interest for the neurophysiologists community. I have only minor comments that should be addressed by the authors prior publication:

·      The authors should explain how they can discriminate the spiking activity from PCs versus other cell types (e.g., GoC, interneurons) in their single unit recordings or discuss to what extent other cells can contaminate the recordings.

·      The authors argue that ‘Climbing fibers influence on MLIs and PCs activity should be negligible in our experiments since PC complex spikes were not detected’. To completely discard the influence of CF input in their experiments, the authors should (1) demonstrate their ability to detect complex spikes through MEAs when they do stimulate CFs and (2) that the spikes that they record are indeed generated in a large part by PCs (see comment#1). Similarly, the authors should consider/discuss the possible contribution of a direct activation of granule cells.

·      In Figures 2, 3A, 5, 6B, 8, 9B, it would be helpful to have the complete outline of the lobules and layers in the recorded slices.

Author Response

In this article, Monteverdi and colleagues investigated the spread of neuronal activity across the cerebellar cortex induced by the stimulation of mossy fibers in the granule cell layer. The analysis of the spatial distribution of single unit recordings using high-throughput high-density multielectrode array revealed an anisotropic and frequency-dependent propagation of the signal with interesting differences according to the orientation of the slice (sagittal versus coronal). The results suggest a role for local inhibition and short-term plasticity in controlling the propagation of the signal, which should impact how cerebellar microcircuits process information.

We thank the Reviewer for the positive comments. We addressed all the issues in the point-by-point response below.

Overall, the article is logically written and the methodological approach is sound. The results bring new and interesting information that will be of interest for the neurophysiologists community. I have only minor comments that should be addressed by the authors prior publication:

  • The authors should explain how they can discriminate the spiking activity from PCs versus other cell types (e.g., GoC, interneurons) in their single unit recordings or discuss to what extent other cells can contaminate the recordings.

A further specification has been added to Methods: “PCs were identified based on three main parameters: i) their location on the slice, between the granular and molecular layers following the lobule layout; ii) their firing frequency, ranging from 10 to more than 100Hz; iii) the spike amplitude exceeding 100μV.” Concerning other cell types with spontaneous activity, Golgi cells are located in the granular layer and stellate cells are located in the external molecular layer, so that these neurons cannot contribute to single-unit PC signals. A possible contamination by basket cells cannot be completely ruled out, given that these cells have their cell bodies located in the internal molecular layer and some can be found near PCs. However, basket cells are smaller than PCs and usually have lower spontaneous discharge frequency, so that their spikes are unlikely to be detected and confused with those of the PCs. Therefore, although we cannot exclude that some basket cells could have been included in the analysis, their impact on the results should be negligible. A sentence has been added to Methods to address this issue (lines 153-162).

  • The authors argue that ‘Climbing fibers influence on MLIs and PCs activity should be negligible in our experiments since PC complex spikes were not detected’. To completely discard the influence of CF input in their experiments, the authors should (1) demonstrate their ability to detect complex spikes through MEAs when they do stimulate CFs and (2) that the spikes that they record are indeed generated in a large part by PCs (see comment#1).

1) We tested the ability of the HD-MEA to detect complex spikes by positioning the stimulating electrode underneath the PC layer. In a subset of channels showing PC activity, we indeed recorded complex spikes in response to single pulse stimulation (see new inset in Fig.2). Since we never observed similar PC responses after mossy fiber stimulation, the climbing fiber contribution in the results reported in the paper is excluded. This is now reported in the text (Methods at line 162-164, Discussion at line 661-664).

2) The explanation required is already included in the response to comment#1.

Similarly, the authors should consider/discuss the possible contribution of a direct activation of granule cells.

Mossy fibers stimulation using a bipolar tungsten electrode is a common procedure in cerebellar research and has been used by this and other laboratories over the last twenty years. A thorough characterization of the effect of this stimulation is reported in Maffei et al., 2002, where specific experiments were conducted to elucidate whether other excitable elements could be directly excited in this condition. These experiments demonstrated that neither granule cells nor Purkinje cells were directly excited by the stimulating electrode. Granule cells were involved in the synaptic response to mossy fiber stimulation. This is evident also in our experiments, since granular layer responses have an average delay of about 1.6ms, compatible with synaptic transmission and not with direct electrical stimulation. The reference Maffei et al. (2002) is now reported in the Methods section with a sentence on the specificity of this stimulation protocol (line 116).

  • In Figures 2, 3A, 5, 6B, 8, 9B, it would be helpful to have the complete outline of the lobules and layers in the recorded slices.

We overlayed the lobules outline to the slices shown in the figures.